# A Holistic Analysis of Food Security Situation of Households Engaged in Land Certification and Sustainable Land Management Programs: South Wello, Ethiopia

**DOI:** 10.3390/foods12183341

**Published:** 2023-09-06

**Authors:** Bichaye Tesfaye, Monica Lengoiboni, Jaap Zevenbergen, Belay Simane

**Affiliations:** 1Centre for Food Security Studies, College of Development Studies, Addis Ababa University, Addis Ababa P.O. Box 1176, Ethiopia; b.t.tessema@utwente.nl; 2Department of Urban and Regional Planning & Geo-Information Management, University of Twente, Languid 1430, P.O. Box 217, 7500 AE Enschede, The Netherlands; j.a.zevenbergen@utwente.nl; 3Center for Environment and Sustainable Development, College of Development Studies, Addis Ababa University, Addis Ababa P.O. Box 1176, Ethiopia; belay.simane@aau.edu.et

**Keywords:** food insecurity, tenure insecurity, determinants, coping, survival, Dessie Zuria, Kutaber, South Wello, Ethiopia

## Abstract

Land degradation, food and tenure insecurity are significant problems in the northern highlands of Ethiopia, particularly in the region known as the country’s famine corridor. Addressing these twine issues in the region has become a focal point for both local and international organizations, underscoring the significance of preventive measures. Since 2000, the Government of Ethiopia (GoE) has been implementing sustainable land management and certification programs. This study aims on households involved in these programs, specifically in Dessie Zuria and Kutaber Woredas, South Wello Zone (SWZ). The primary objectives of the research were to assess households’ current food security status, identify factors influencing their food security, and classify coping and survival strategies employed by households during food shortages. Primary and secondary sources have been used to collect both qualitative and quantitative data. Quantitative data were collected from surveyed households and analyzed USING SPSS software version 26, whereas qualitative data were transcribed, grouped, and interpreted in line with the aim of the research. Three food security models, namely the Household Food Balance Model, Months of Adequate Household Food Provisioning, and Household Dietary Diversity Score, were employed to evaluate food security. Consequently, a significant percentage of the surveyed households, amounting to 88.3%, 35.6%, and 93.8%, were found to experience food insecurity according to the respective models. Rainfall shortages and variability, crop pests and diseases, shrinking farm plots, and land degradation are among the identified food security determinants. During dearth periods, households deploy a variety of coping and survival strategies. To mitigate food insecurity stemming from both natural and socio-economic factors, the research suggests several recommendations. These include advocating for tenure policy reforms by the GoE, and the local governments should promote the adoption of efficient land management practices, instituting a land certification system based on cadasters, encouraging family planning, boosting investments in education and literacy, raising awareness and providing training in climate-smart agriculture techniques, educating communities on optimal grain utilization, saving, trade, and storage methods, facilitating opportunities for income generation through off-farm and non-farm activities, and offering support for crop and livestock diversification.

## 1. Introduction

In an ever-evolving world, achieving food security is confronted with increasingly intricate challenges. The ongoing battle against poverty and the mission to ensure food and nutrition security while safeguarding the environment remain significant obstacle for international development initiatives [1,2]. Globally, between 702 and 828 million people have experienced hunger in the world in 2021 [3]. The overwhelming impacts of food insecurity are most pronounced in Sub-Saharan Africa (SSA), where a substantial portion of its population grapples with persistent hunger and malnutrition. Over the last quarter-century, the agricultural growth rate in sub-Saharan African nations has remained stagnant at three percent [4]. Despite efforts to improve food security, the situation in SSA remains precarious. Along with this, the COVID-19 pandemic has further fueled and worsened the region’s food insecurity, shift in consumer demand, and disruption of the supply chain with lockdowns and restrictions affecting agricultural production, transportation, and trade [5,6]. The inability to move freely, the demand for physical separation to keep people safe, as well as the need for additional personal protective equipment diminish the effectiveness of many businesses [7]. The COVID-19 pandemic, which has rapidly and extensively spread worldwide since late 2019, has had far-reaching consequences on the socioeconomic conditions of individuals, particularly in relation to food security and nutrition [8]. Furthermore, the ongoing Ukraine-Russia war has affected SSA’s food security by potentially raising global food prices through disruptions in Ukraine’s agricultural production and exports. Ukraine and Russia are significant exporters of wheat, corn, and barley, and disorder in their output resulted in higher global food prices [9].

Ethiopia is the second-most populated country in SSA [10]. Agriculture is the mainstay of the economy and the primary source of employment and income for more than 85 percent of the population. Nevertheless, the sector, which is sensitive to various shocks, trends, and seasons, cannot feed the country’s vast population [11]. The demand for humanitarian aid in Ethiopia has surged significantly since 2019 due to a combination of factors, including drought, security concerns, and the ongoing crisis in northern Ethiopia. It is estimated that around 20 million individuals may necessitate humanitarian assistance in the year 2022 [12]. In addition, the Africa Food Security and Hunger Indicator Scorecard indicates, Ethiopia has the highest number of people in undernourishment/hunger, affecting 32.1 million people, which makes it the fourth African country scoring 37 percent in the number of undernourished people [13]. Ethiopia’s root causes of food insecurity include, not limited to, unpredictable weather patterns, land degradation, lack of tenure security, rain-fed agriculture, and low agricultural productivity [3,14]. The issue of land tenure is also a significant challenge in Ethiopia, particularly in rural areas. Land is primarily owned and controlled by the government, and farmers often lack secure land tenure rights, which limits their interest in investing in land-augmenting practices and improving land productivity [15]. 

The Ethiopian Government has initiated extensive programs for land certification and sustainable land management, aiming to enhance agricultural practices, establish secure land tenure for small-scale farmers, and encourage investments in land management endeavors [12,16]. 

Several national and international organizations and GoE have collaborated to materialize the three twine development pillars: food security, land management, and land certification. However, there is still a long way to go in addressing Ethiopia’s complex food security and land situation, particularly amid the ongoing climate change and rampant conflict.

The issue of food security, land certification, and sustainable land management has been subject to several studies in Ethiopia and abroad. Among the notable works, secure land tenure and sustainable land management (SLM) [17]; Contribution of land registration and SLM [18]; Impacts of land certification on tenure security, land dispute, land management and agricultural production [19]; land tenure reforms, tenure security, and food security in poor agrarian economies and causal linkages [17]; food security in Ethiopia [20], are to mention a few. These studies did not address the back-and-forth linkage of the three variables. Since then, several intrusions and uptakes have been undergone. This intervention could improve the food security situation of rural households as well as land management and tenure security via recent strategies and policies, and there is still room for improvement in the food security of households. Furthermore, as recommended by Saguye [21], such improvement should be supported by site-specific empirical data. Therefore, this paper aims to evaluate the degree of food security among rural households participating in SLM and land certification programs. The study aims to achieve the following specific objectives: (i) assess the food security status of the households under scrutiny; (ii) identify the factors that exert an influence on household food security; and (iii) classify the prevalent coping and survival strategies that households employ in times of food scarcity.

The finding of this research could be an input for policymakers, planners, and other interested parties who wish to intervene and improve the country’s food security and land-related conditions. Seven parts make up the paper’s subdivisions. Following this introduction, Section 2 gives a literature review, Section 3 focuses on the materials and procedures, and Section 4 delivers the results. Section 5 contained a discussion. Section 6 focuses on the conclusion, and Section 7 provides recommendations. 

## 2. Related Literature

This chapter presents contemporary ideas and concepts pertinent to the emerging topics related to the research. It elaborates the conceptual definition of food security, coping, survival strategies, and theories of food security. Additionally, it briefly provides a conceptual explanation of land degradation and sustainable land management, tenure security, and the interplay between the three core variables of the study.

### 2.1. Food Security: The Concept

Food security as a concept first emerged over 50 years ago during the 1970s’ extraordinary volatility in agricultural commodity prices, which followed the chaos in the currency and energy markets [22]. The concept’s ongoing evolution reflects a growing recognition of its intricacies in research and public policy. Initially, food security as a discipline focused on assuring food supply in terms of availability and the stability of the prices of staple commodities on a local and global scale. Since its emergence as a dynamic concept, it has consistently incorporated new aspects and dimensions over time [22].

The main issue that the idea of food security is now facing is having a suitable and practical definition [23,24,25,26]. After passing through difficulties seeking a proper description of the concept and bringing greater coherence to such complexity, a new definition of food security was put through international dialogues at the 1996 World Food Summit [27]:


*“Food security, at the individual, household, national, regional, and global levels, is achieved when all people, at all times, have physical and economic access to sufficient, safe, and nutritious food to meet their dietary needs and food preferences for an active and healthy life.”*


The 1996 World Food Summit came with pillars of food security, which, in practice, are assumed to be a foundation block to evaluate and measure the prevalence of food security in temporal and spatial dimensions. In the meantime, these pillars, or intricate determinants of food security, (1) availability, (2) access, and (3) utilization, are deliberately proved to give equivalent meanings for terms in definition and to measure the extent and magnitudes of food in/security. 

The 2009 World Summit on Food Security introduced the fourth pillar, stability, referring to consistency in food security [28]. The four pillars are undoubtedly linked and dependent on one another rather than static and distinct; therefore, the pillars’ portrayal of the notion is deceptive [28]. Pillars do not illustrate the relationship between the aspects of food security. The weighing problem [29,30] is another objection to the idea that food security depends on four “pillars”. Contrary to the pillar analogy, not all aspects of food security are equally crucial.

#### 2.1.1. Major Theories of Food Security

Defining food insecurity’s underlying and predisposing causes requires relevant theories that best address the problems. It appears difficult to accurately define “theory” as a notion in a way that is widely accepted. As a result, authors would instead list its broad characteristics so that their readers may choose one or a mixture of them that might work for their own unique needs. The diverse schools of thought orchestrate different theories of food security. However, the field has no such a full-fledged and stand-alone theory. The food security line of thought has borrowed them from disciplines like economics, biology, and political science. The ‘political explanation’ or ‘general explanation’ and ‘the two modern famine theories’ are chief among these theories. 

Accordingly, Hussein [31] also shows that using a single theory to study every element of food security is challenging. With this perspective in mind, this study applied the two most popular theoretical underpinnings that could complement one another to represent a trustworthy image of household food security and its nexus to land certification and sustainable land management.

Political Economy Explanations: According to advocates of this theory, socio-economic and political instability, government failure, unequal access to resources, war, civil unrest, and ecological degradation are the root causes of food insecurity [32]. The proponent of this theory links the drought-induced famine in Western Sudan to ecological degradation. In particular, the Sahara Desert’s conversion of arable land and the exposure of vulnerable populations to famine were linked to the failure of the then government to serve the entire nation democratically. The increasing disparity between pastoralists and other segments of the regional and national economy and within the pastoralist community resulted in social differentiation and remained an element of famine causation [33]. 

Food Availability Decline Model: The Food Availability Decline (FAD) model seeks to explain the major hindrances to increasing agricultural output, which in turn causes a drop in food availability. The main contention of this theory is that anything that interferes with agricultural output, such as drought, land degradation, and flooding, can lower the amount of food available for a lengthy period and result in famine [34]. According to this school of thought, food production is contested using numerous factors: population growth, diminishing per capita livelihood resources, fragmentation and competition over the resource, and natural and human-induced hazards like drought, land degradation, flood, rainfall shortage, and variability, crop pest, and livestock disease.

Among the prominent scholars who criticized this theory in this regard is Sen. According to him, a person cannot be entitled to consume food just because it is available in the market or in the economy, and famine can happen even if the overall supply remains stable. Sen offers a variety of examples to support his claim, including the Bangladesh famine of 1974, the 1973 and 1984 Ethiopian famines, and the Bengal famine of 1943 [35]. Though exposed to critics, Sen has produced a new model of famine explanation called Food Entitlement Decline (FED). The theory implies that human food needs are fulfilled using means of entitlements rather than production or the availability pillar alone. The main weaknesses of the FED model include failure to consider the intra-household distribution of food and exclusion of relief entitlement.

#### 2.1.2. Determinants of Food Security

Studies have been undertaken on the factors that affect food security concerning various levels, regions, and times. According Kim [36], household-level food security is influenced by per capita landholding, livestock availability, education level of the household head, farm and non-farm income, soil fertility, and conflicts. Parallel empirical research conducted in Ethiopia’s North Wello using the Food Balance Sheet has revealed almost identical elements as a drivers household food security [37].

In a study done in Sekyere Afram District of Ghana by Aidoo et al. [38], age, gender, marital status, household size, off-farm income, credit access, and remittance were determinants of household food security. Additionally, K. Tesfaye [39], in his study of household coping strategies and policy options in the Arsi zone, Dodota—Sire Woreda, listed family size, number of oxen owned, use of chemical fertilizer, size of cultivated land, credit, consumption expenditure, livestock owned, and off-farm income were significant determinants of household food in/security. Similar research on determinants of household food security by Ngema et al. [40] in South Africa listed household size, age, income, gender, marital status, education, access to credit, livestock farming, irrigation, and participation in the “*one home, one garden program*”.

#### 2.1.3. Coping and Survival Strategies

According to S. Maxwell [41], coping strategies are a collection of acts a household does in a specific order in reaction to shocks like hunger, drought, and other natural and human induced calamities. Households who struggle with food insecurity do not just accept these bad circumstances; they actively work to improve them [42]. People who leave their communities in times of hunger or starvation in search of a job and food should not be viewed as passive victims but as losers in a valiant battle for existence. Therefore, when food shortage happen, individuals attempt to manage and do not rely heavily on outsiders unless everything else gets beyond their control. Differentiating between a “coping strategy” and an “adaptive strategy” frequently sounds onerous. However, scholars in the field can distinguish between these two groups of strategies depending on the anticipated outcomes and temporal and spatial dimensions. Longer-term adaptive methods help individuals react to a new set of changing situations that they have not encountered before. Adaptation may be autonomous and planned. Over time, individuals have developed coping mechanisms in response to their lifelong experience in dealing with familiar and predicted changes in seasons, as well as their responses to the evolving characteristics of each season [43]. These strategies encompass measures such as enhancing production and productivity, acquiring grains, selling livestock, engaging in daily wage labor, and participating in small-scale trading [44]. The specific conditions at the local level shape the coping and survival strategies to be employed, and reflects individuals’ resourcefulness and adaptation to familiar and predicted seasonal changes.

### 2.2. Sustainable Land Management

The concept of SLM originated from the 1992 Earth Summit and was first utilized by [45]. The World Bank’s definition of SLM focused on the responsible use of land resources to meet present and future needs while maintaining or enhancing the land’s productivity, ecosystem services, and resilience. The bank emphasized the integration of economic, social, and environmental considerations in land management practices [46]. The fundamental principle underlying SLM may appear straightforward initially. However, it stands for one of the most ambitious aims in real-life scenarios, aiming to achieve sustainable use of natural resources [47]. 

At the local level, factors such as rapid population growth, overgrazing, climate change, land degradation, insecure land tenure, and improper agricultural practices have worsened the problem of land degradation. These issues could become significant barriers to achieving Sustainable Development Goals [48]. The dilemma of reversing ecosystem degradation while fulfilling the increasing need for their services can be partially tackled. However, successfully adopting SLM necessitates significant transformations in policies, institutions, and practices that are presently not implemented [49]. Sustainable land management aims to ensure sufficient current production levels while preserving the land resource base for future generations, thereby safeguarding development opportunities that seek to achieve a balanced transformation in the social, ecological, and environmental dimensions of human well-being [50]. 

Factors that restrict the rate of acceptance and usefulness of each SWC practice depend on the degree and intensity of the problem, slope and terrain, agricultural system, and other socio-economic and institutional settings [51]. According to Tefera and Sterk [52], the availability of technical assistance, the structure’s suitability for ongoing agricultural operations, the need for short-term practice, labor demand, tenure security, and slope are a few determinants for selecting specefic SWC practice.

### 2.3. Tenure Security

In the highly competitive pursuit of resources, ensuring property rights and tenure security becomes crucial pillars of the global development agenda. There is a pressing need for a clear understanding of the connections between tenure security and sustainable development in our world [53]. According to the same source, SDG 1.4.2 also refers to the perception of certain rights and thus qualifies the point about legal documentation.

The importance of tenure security extends to various areas. It impacts various levels, which explains why it has attracted the interest of environmentalists, ecologists, climate scientists, women’s empowerment advocates, food security specialists, public health practitioners, and other individuals and groups concerned [54]. Conceptual notions support that a secure land tenure can contribute to establishing a relationship between land tenure and food security [55]. Land tenure security is a topic of considerable debate regarding its significance for impoverished populations in rural and urban areas of developing nations [56]. According to Larson [57], tenure security has received global attention due to its significance as an enabling condition for social and economic development. Since the emergence of the concept, there have been multiple definitions and interpretations of tenure security [58,59,60,61,62].

Most importantly, Roth et al. [62] defined land tenure security based on individuals’ perceptions regarding the certainty of land rights. 


*“Land tenure security exists when an individual or group is confident that they have rights to a piece of land on a long-term basis, protected from dispossession by outsider sources, and with the ability to reap the benefits of labor and capital invested in the land, whether through direct use or upon transfer to another holder.”*


The definition provides an extensive explanation of land tenure security, encompassing three fundamental elements: breadth, duration, and assurance. Notably, this definition applies to all forms of land, including residential and agricultural land, thereby allowing for meaningful comparisons with other definitions in the field. Nonetheless, while there is limited consensus regarding the precise definition of tenure security, even less agreement exists on how the several types of conceptualizations of tenure security may be interconnected [63].

The attainment of downstream goals, such as conflict management and resolution, food security and economic growth, and gender equality, is seen as being influenced by, if not a condition for, land tenure security [64]. According to Maxwell and Wiebe [65], land tenure is managed using social relationships and institutions (formal, statutory, legal, and local/customary) that decide access to, use of, and transfer of the land that is part of a bundle of rights available to diverse types of individuals [66].

Land security in a formal manner is achieved via land certification. Land certification provides tenure security to the owner and prevents, in most cases, from entering disputes. Insecurity in land tenure can arise from various sources, both within formalized and informal arrangements. One primary cause of tenure insecurity is the lack of alignment between legally recognized rights (de jure) and the actual rights exercised in practice (de facto). This disparity, often called the “tenure gap,” can give rise to misunderstandings, and disputes, which in themselves can have significant implications [67]. Land resource is often a source of dispute. As reported by Bob [68], conflict can happen between households, neighborhoods, and neighboring communities over land rights and boundaries. Establishing a functional and widely accepted tenure system is crucial to sustain such misunderstandings. Resolving any ambiguities in the land tenure system should take precedence even before the commencement of adjudication [69].

## 3. Research Methods

### 3.1. Description of the Study Area

The research centers on South Wello Zone (SWZ), which is one of the zones within the Amhara National regional state (ANRNS). There are multiple reasons behind selecting SWZ; despite its agricultural potential, like other regions in Ethiopia, it faces considerable challenges in ensuring food security. Land degradation is also a pressing issue, particularly in areas with steep slopes and poor land management practices prevails. Furthermore, since 2000, the zone has been the focus of land certification and SLM programs.

South Wello (Figure 1c) borders regions and zones. The North Wello Zone is to the north, while the Oromia Special Zone lies to the east. The Afar Region is located to the northeast, and in the southeast is the Argobba special Woreda. It has a total area of 17,067.5 square kilometers and is home to an estimated population of 3,239,475 people [70], with a population density of 170 persons per square kilometer [71].

Two Woredas, Dessie Zuria and Kutaber, were selected based on their vulnerability to land degradation and food insecurity and the existence of SWC and land certification programs (Figure 1d). Survey sites have been chosen from six Kebeles, with three selected from each Woreda, and these sites represent the three agroecological zones: Dega, Woynadega, and Kola [72]. 

### 3.2. State of Agriculture in the Study Area

The study area has diverse agroecology, including Dega (cool and humid), Woynadega (Cool sub-humid), and Kola (warm semi-arid) [71]. The most common soil types found include vertisols, cambisols, and luvisols, which are conducive to growing various crops [70]. The rainy season is from June to September, with an average annual rainfall ranging from 800 mm to 1200 mm, and experiences high inter-annual variability in rainfall, which can impact crop yields and livelihoods [73]. Agricultural in the study area is a mixed cereal-livestock system. Cereals hold a dominant position, accounting for more than 73 percent of cultivated crops, followed by pulses at 24 percent, while the remaining three percent is allocated to oilseeds [74]. Among the cereal varieties, teff (*Eragrostis tef*), sorghum (*Sorghum* spp.), wheat (*Triticum vulgare*), and barley (*Hordeum vulgaris*) occupy prominent roles, constituting 25, 18, 16, and 11 percent of the total, respectively [74].

The crop cycle aligns with a bimodal rainfall pattern, featuring a short spring (Belge) and a longer summer rainy season (Meher). This leads to two distinct harvest periods, with the spring harvest assuming paramount importance. This significance arises due to the rugged topography and intense summer rains, enabling cultivation on gently sloping areas, albeit covering a small land area. Additionally, certain lowland villages capitalize on spring rains to cultivate teff. These rains are also of significance for mid-highland farmers cultivating long-maturing varieties of tef, sorghum, barley and maize. Crop production thus serves as the primary income source for households. 

Similarly, rural households in this region practice a mixed agriculture approach, concurrently engaging in both crop cultivation and livestock rearing. Livestock serves as the driving force for draught power, while household members contribute the necessary labor for farming operations. The livestock portfolio includes cattle, sheep, goats, pack animals, and poultry [75].

### 3.3. Materials and Methods

#### 3.3.1. Data Set and Analysis Method

Academics have long acknowledged the need for diverse assessment methods due to the multifaceted character of food security [76]. The “holy grail” of evaluating food security would be developing a standard indicator that captures all aspects while being valid, trustworthy, similar across time, and comparable across locations. Despite the emergence of diverse indicators during the previous ten years, none of them found to satisfy these requirements [77]. The research applied mixed method research that require the application of both qualitative and quantitative data from primary and secondary sources. Both quantitative and qualitative assumptions were made when they engaged in their research (Creswell, 2009:10). The research was conducted by utilizing data obtained via a survey administered to 371 farm household heads selected via a multi-stage sampling technique [74]. The study targeted farmers participating in land certification and SLM within the Dessie Zuria and Kutaber Woredas regions. A sampling frame, serving as a list identifying the target population, was derived from six chosen kebele administration offices. 

Data for the research originated from both primary and secondary sources. Primary data were gathered using methods including household questionnaires, focus group discussions, key informant interviews, personal field observations, and community workshops (Table 1). Complementary to this, secondary data were sourced from pertinent local authority reports, books, and journals. The questionnaire responses from participants were subjected to quantitative analysis, resulting in their summarization and presentation using tables, graphs, and percentages. The questionnaire encompassed both closed-ended and open-ended inquiries.

Data collection took place between January and February 2020, a suitable period when farmers had completed their harvest activities and initiated natural resource development campaigns at the watershed level. This timing facilitated the ease of interviewing sample farm households and collecting necessary data. Prior to the survey, the questionnaire was pre-tested on one percent of the total sampled households. Feedback gathered from farmers during this phase prompted adjustments to ensure the questionnaire’s reliability and validity.

Additionally, secondary data were acquired from various sources such as books, journals, and internet repositories. The study employed three models—Household Food Balance Model (HHFBM), Household Dietary Diversity Score (HDDS), and Months of Adequate Household Food Provisioning (MAHFP).

The analysis of the data was performed using both SPSS and software version 26. For the quantitative data, a range of techniques was employed, including the creation of frequency tables, calculation of percentages, generation of graphs and figures, cross-tabulation, and utilization of diverse descriptive statistical approaches. Concurrently, qualitative data underwent processes of transcription, categorization, and examination aligned with the predefined research objectives.

#### 3.3.2. Food Security Measurement Tools

Researchers focusing on food security recommend the application of instruments that thoroughly assess the four pillars of food security. India provided one of the earliest examples of home food security monitoring; the British administration created the Indian code in 1880 due to India’s prolonged famines [77]. No globally recognized indicators are used as measuring instruments, so assessing food insecurity is still a complicated subject [78]. Establishing reliable methods for measuring food insecurity is one of the primary steps crucial to addressing this pressing issue. Without accurate measurements, it becomes impossible to target interventions effectively, monitor and evaluate programs and policies, or derive valuable lessons to enhance the efficacy of measures in the future. These recommendations led us to the use of three models/tools listed in the subsequent section.

##### Household Food Balance Model

The Household Food Balance Model (HHFBM) assesses the food availability pillar of food security in the household. The Household Food Balance Model is a straightforward equation modified from the FAO Regional Food Balance sheet [79]. The model was also utilized by other researchers to examine the state of food security in several areas of Ethiopia, such as [80,81,82,83]. It calculates the net amount of food grain each rural household possessed over a year. The availability of food at the home level is often determined using the formula below: NGA = (GP + GB + GO) − (HL + GR + GS + GG)
where NGA = Total Grain Available/year/household.
GB = Total Grain Bought/year/household
GP = Total Grain Produced/year/household 
GO = Total Grain Obtained/year/household, 
HL = Total Post Harvest Loss/year/household 
GR = Amount of Seed Reserved for Seed/year/households
GS = Amount of Grain Sold/year/household
GG = Grain Given to Others/year/household.

The HHFBM analysis variables involved four main steps. Firstly, the net grain available for each household in kilograms (NGA) is converted into total kilocalories using Ethiopia’s specific conversion factors [84]. Secondly, the household-level food supply is calculated by dividing the total number of days in a year by the adult equivalent value for each sampled household, resulting in the total available calories per adult equivalent per day. Thirdly, the minimum calorie requirement for an adult equivalent per day was set at 2100 kilocalories, based on Ethiopia’s 1996 Food Security Strategy (FSS). This value represents the number of calories necessary for an adult to maintain a healthy and moderately active lifestyle. Lastly, a comparison was made between the available kilocalories (supply) and the required kilocalories (demand). All the necessary data for the HHFBM model were obtained from respondents covering the period between December 2021 and January 2022. According to the HHFBM analysis results, a family is considered food secure if the per capita calorie availability is equal to or greater than 2100 Kcal. A value between 1750 and 2100 Kcal denotes a mild food insecure situation, between 1500 and 1750 Kcal denotes a moderately food insecure, and below 1500 Kcal shows the severity of food insecurity.

##### Household Dietary Diversity Score

The Household Dietary Diversity Scale (HDDS) examines the utilization pillar of food security by counting how many different food types of each household consumed over the past 24 h. The food groups included in the HDDS are intended to stand for a variety of foods, from those that do not contribute to a nutritious diet but cost money to buy, like sugar, sweets, beverages, and condiments, to foods that improve the quality of the diet by supplying essential nutrients [85]. These later foods include grains, fruits, vegetables, fats, oils, and protein from plants and animals sources. 

During the survey we questioned respondents about the meals consumed by any family member in the past 24 h. The tool consisted of questions regarding sixteen food categories, after aggregated into twelve categories for analysis. The score is a straightforward, adding the twelve food categories that each family member has ingested [86]. Based on the total household distribution, terciles were scored for dietary diversity (DD). Low DD is defined as three or fewer food categories, medium DD as four, and high DD as five or more food items.

##### Months of Adequate Household Food Provisioning

The Food and Nutrition Technical Assistance program of the United States Agency for International Development (USAID) developed the tool Months of Adequate Household Food Provisioning (MAHFP). The tool measures the access pillar of food security and counts how many months a particular household has had enough food. 

One household member is given this questionnaire and responds on behalf of the rest family members. Any adult household member (ideally the household head) who is over the age of 18 may respond to these survey questions. According to Swindale, Bilinsky [87], these questions aim to identify the months in which there is limited access to food, irrespective of the food source. 

Households were divided into four categories of food /security: ≥12 months food secure, ≥10–12 months mildly food insecure, ≥7–9 months moderately food insecure, and ≤6 months severely food insecure [88].

## 4. Results

The outcomes presented in this section have been obtained through diverse data collection techniques and analytical methods, and they will be expanded upon in the upcoming subsections. In Section 4.1, the socio-economic characteristics of the respondents, including their sex, age, education, and wealth, will be described to provide an understanding of the demographic profile of the households under investigation. Building upon this, Section 4.2 delved into assessing the food security situation of the studied households, exploring aspects such as food access, availability, and stability to provide a comprehensive overview of their food security status. Section 4.3 focused on identifying the determinants of food security, and Section 4.4 investigates major coping and survival strategies deployed by households during food shortages.

### 4.1. Respondent’s Socio-Economic Characteristics 

In the study, there were a total of 371 respondents participating in land certification and SLM programs. The majority of these households, constituting 80 percent, were led by male household heads. In contrast, approximately one-fifth, or 20 percent, were headed by women. The age of the respondents spanned from 20 to 71 years. Notably, around 60 percent of the respondents fell into the 31–40 and 41–50 age categories, indicating that farming was the primary occupation for individuals in these age groups compared to others. The age distribution pattern indicated that a significant 90 percent of the population fell into an age group that represented a potential workforce capable of participating in development initiatives like watershed management. Regarding educational background, a substantial portion of the respondents, accounting for 58 percent, lacked formal education and were not literate. Conversely, only 42 percent of the respondents were literate, which posed challenges for introducing innovations and new agricultural practices in a predominantly illiterate community.

Regarding the wealth category formulation, many factors were considered. During the focus group discussions (FGD), community members asked how wealth groups were defined. Farm size, the number of oxen and shoats, ownership of transport animals, labor availability, and owning a flour mill were among the parameters considered. Only a tiny percentage of respondents, 1.75 percent, found themselves as better-off households, while the majority, 98.25 percent, considered themselves either poor or middle. High proportion of poor households seen in the study could be attributed to factors such as land degradation, shrinking farm sizes, and poor agricultural productivity, which further exacerbates wealth disparities among community members.

In two Woredas, two rounds of land certification programs were executed. According to data from the South Wello Land Use and Land Administration office (SWLULAO), a total of 947,474 land parcels underwent certification, providing benefits to 81,412 households. Additionally, the implementation of various soil and water conservation practices (SWCPs) in the South Wello Zone (SWZ) has exhibited substantial expansion, with the area covered growing from 2478 hectares in 2000 to 6607 hectares by the year 2020 [87].

### 4.2. Food Security Situation of Households

The study examined the food security status of households by employing three different food security models. The outcomes derived from each model are detailed as follows: 

#### 4.2.1. Household Food Balance Model

Through a comparison of the total food supply and the total consumption, this model provides insights into whether the household is food-secure or not. Based on the information provided, HHFBM is used to measure the availability aspect of food security. After implementing various interventions to enhance agricultural productivity, the study area was assumed to be a surplus producer. However, despite these assumptions, the research reveals a different reality.

Among the households included in the survey, the majority, totaling 88.3 percent, were categorized as food-insecure (Table 2). Within the group of food-insecure households, 69.7 percent experienced severe food insecurity, 11.45 percent faced moderate food insecurity, and 7.2 percent encountered mild food insecurity. These percentages underscore the significant levels of food insecurity prevalent among the surveyed households, with a substantial portion experiencing severe food insecurity. Consequently, farmers encounter difficulties in accessing a consistent and sufficient supply of high-quality food to meet their dietary needs. Conversely, the remaining 11.7 percent of households were classified as food-secure, indicating that they had an adequate food supply to fulfill their nutritional requirements. 

The findings indicated a wide range of daily calorie consumption per household, with the lowest recorded at 78.1 kilocalories (Kcal). This situation underscores that a significant number of households are grappling with severe food shortages, which can have adverse effects on their health and well-being. In contrast, the highest daily calorie consumption observed was a substantial 7543.3 Kcal per household, signifying that certain households had access to an abundance of food that exceeded their nutritional needs. The average daily calorie consumption per household, calculated at 1272.1 Kcal, offers a general overview of the overall food security situation among the surveyed population. It’s important to note that extreme values, both low and high-calorie intake, can significantly influence this average. The standard deviation, which stands at 865.7 Kcal, indicates the degree of variability in calorie consumption among households. A higher standard deviation implies greater disparities in food availability and consumption patterns within the population. In summary, the results obtained from HHFBM reflect the diversity and complexity of the food security situation among the surveyed households. These findings reveal a spectrum of calorie intake levels, ranging from critically low to excessively high, with an average intake falling within a specific range.

#### 4.2.2. Household Dietary Diversity Score

The research also utilized the HDDS tool to assess the food security situation of households under study. HDDS focuses on measuring the variety of food items consumed by household members within the last 24 h and intended to measure the utilization pillar of food security (Table 3).

According to HDDS model, out of the total number of respondents, 239, or 64.4 percent of households, reported consuming five or more food items. This shows an elevated level of dietary diversity among these households. Furthermore, the table reveals that 12 percent of households fall under the medium dietary diversity (DD) category, while 23.6 percent fall under the low DD category (Table 3). These percentages show that a sizeable proportion of households have a limited range of food items in their diets. 

The research suggests that the higher DD score seen could be attributed to the engagement of households in irrigation programs. These programs enable households to produce and consume diverse types of vegetables, thereby increasing their dietary diversity. Additionally, households taking part in the Productive Safety Net Program (PSNP) through labor receive monthly cash, wheat, and oil. This inclusion of wheat and oil in their household’s food, along with the cash assistance, allows them to access and purchase additional food items, contributing to their higher DD scores. Moreover, the study area, particularly the two specific Woredas, is known for having a substantial livestock population. The opportunity to include livestock and livestock products in the dietary mix has contributed significantly to the observed dietary diversity among households in the area. Overall, the HDDS indicates that while a sizeable portion of households have achieved high dietary diversity, there are still households with medium and low DD scores, highlighting the need for further interventions to improve food security and diversify diets in those households.

#### 4.2.3. Months of Adequate Household Food Provisioning

MAHFP, which assesses the adequacy of a household’s food access throughout the year, takes into account both the quantity and quality of available food. It calculates the number of months during which a household’s food supply satisfies its nutritional requirements, providing an estimation of long-term food security.

Based on this model, it can be deduced that 93.8 percent of the households surveyed in the study are grappling with precarious food insecurity (as shown in Table 4). This implies that the majority of these households do not have consistent access to a sufficient food supply throughout the year.

Further examination of the data reveals that 30.4 percent of the surveyed individuals fall into the mild food insecurity category, indicating occasional concerns about their ability to access an adequate food supply. Another 44.8 percent of respondents are categorized as experiencing moderate food insecurity, signifying that they frequently face limitations or uncertainty in accessing food. Additionally, 18.6 percent of households are classified as confronting severe food insecurity, suggesting a significant compromise in their access to food, potentially leading to hunger and malnutrition.

In contrast, the study identified a small portion of households, specifically 6.2 percent, as being food secure. These households can provide food for their family members throughout the entire year. They may even have surplus grain that can be utilized for various purposes, such as selling, exchanging, giving, or fulfilling social obligations.

However, in addition to the descriptive analysis, the study also delved into the food situation of households through Key Informant Interviews (KII) and Focus Group Discussions (FGD) panels. Through these panels, our research identified the specific months during which households experience food shortages. The participants in these discussions categorized the year into three phases related to food security: months of food security, months of transitory food insecurity, and months of chronic food insecurity (as illustrated in Figure 2).

Based on the gathered information, food security in the study area exhibits recurring fluctuations throughout the year. Respondents indicate that sustained food security typically commences with the arrival of the new harvest, usually occurring between November and December, and lasts until March. During this period, households experience an increase in grain consumption for various reasons. After the harvest, households allocate a significant portion of their grain for activities such as weeding, charitable donations (sadaqa), religious ceremonies, and celebrations. Additionally, some grain is sold to meet essential household needs and expenses. These practices result in a reduction in the overall quantity of grain stored by these households.

Around March, households enter a temporary phase of food insecurity and employ strategies to balance the availability of grain with consumption patterns. During this period, they make efforts to manage their limited food resources until the next harvest. The period of chronic food insecurity typically begins in June and extends until August or September in various locations. In the Kola region, due to temperature and crop characteristics, individuals navigate the transitional food security phase by consuming quickly maturing crops. In the Dega agro-ecological zone, residents transition through temporary food insecurity from March to June by consuming a yield of rapidly maturing small rainy season barley variety known as Ginbote.

These observations underscore the intricate interplay of multiple factors that contribute to the recurring food security pattern in the research area. Factors such as the timing of harvests, cultural traditions, inefficient grain management practices, weather conditions, and crop choices all influence the fluctuating stages of food security throughout the year. These findings continue to highlight the disparity between the assumed surplus in production and the actual food security conditions on the ground. It is obvious that, despite initiatives like SLM and land certification programs aimed at boosting agricultural output, a substantial number of households in the study area continue to contend with persistent food insecurity.

### 4.3. Determinants of Household Food Security

During a panel discussion, participants listed numerous factors that play a pivotal role in determining food security in their region. These factors encompassed crop pests and diseases, floods, land degradation, locust infestations, diminishing farmland, irregularities in rainfall patterns, scarcity of agricultural inputs, land erosion, a shortage of working oxen, and tenure insecurity.

Drawing insights from the inputs provided by the panelists, the household survey was intentionally designed to incorporate these factors and assess their relative importance. Significantly, it was revealed that a shortage and variability in rainfall received the highest score, accumulating a total of 1691 points, and were identified as the primary impediments to agricultural production, as depicted in Figure 3. In the study area, agriculture heavily relies on rainfall, and inconsistencies in its timing, intensity, amount, and distribution have been recognized as factors that detrimentally affect crops and diminish overall production, leading to food insecurity. Variability in rainfall also impacts farmers’ livelihoods by causing problems such as flooding and waterlogging of crops, again further exacerbating food insecurity. Although irrigation has been introduced in certain areas over the past two decades, it remains at an early stage of development and has not yet made a significant contribution to crop production.

Second in the queue, with a score of 1213, is the prevalence of crop pests and diseases. Many crop pests and diseases are common in the area. Wheat rest, bull worm, aphids, and many fungus species were to mention a few. According to the respondents, since 2010, a new gall-forming disease called Dimereche (in Amharic) has emerged, specifically affecting the Faba bean. This disease has resulted a significant yield loss on Faba beans, crucial for stew preparation and serving as a cash crop for farming households. 

Farmers perceived the shrinking land size as a significant challenge that threatened their ability to sustain their food security. Farm plot shrinkage can be linked with population growth and the intent to share the household’s tiny plots with the newlywed are common. Farmers are concerned about the implications of shrinking land size, including decreased crop production, limited grazing areas for livestock, and the inability to implement sustainable farming practices. Having 1174 points, land degradation stood fourth on the list. Farmers perceive land degradation as a growing concern that threatens their livelihoods and the long-term sustainability of agriculture. They observe the gradual deterioration of their land’s quality, characterized by soil erosion, reduced fertility, water scarcity, and flood. These changes are often attributed to improper farming practices, climate change, deforestation, and improper land management. Farmers recognize the negative impacts of land degradation on crop yields, livestock productivity, and overall farm profitability. The insights garnered from the Focus Group Discussions (FGDs) distinctly illuminate a significant issue: the absence of secure land tenure has emerged as a substantial deterrent impacting farmers’ inclination to engage in Soil and Water Conservation Practices (SWCPs). This hesitancy is intricately linked to the would be land distribution efforts by the government, which has instilled concerns among farmers about the potential loss of their conserved lands due to evolving land policies. This apprehension was particularly pronounced among those who owned land holdings exceeding the average size.

It is important to note that refraining from investing in SWCPs could have noteworthy consequences, particularly in terms of compromising soil fertility and exacerbating land degradation. These discussions have revealed that certain agricultural plots are inherently insufficient in yielding the amount of grain required to meet annual consumption needs. This predicament poses a tangible challenge to ensuring consistent food security within the community.

These findings underscore the intricate web of factors connecting land tenure security, investment decisions, and ecological preservation. The concerns expressed by farmers highlight the complex interplay between socio-economic factors and policy considerations that profoundly influence both agricultural practices and land management strategies.

### 4.4. Coping and Survival Strategies

The research has investigated the coping and survival strategies employed by households living in six survey kebeles during KII and FGD platforms. The studied community has a long history of proving its ability to adapt to harsh environmental conditions. This study has uncovered a wide range of household coping mechanisms for the vicissitudes of food insecurity brought on by environmental and human-induced disasters.

In the early stages of food shortages, households adopted a primary strategy, as depicted in Figure 4. Among the surveyed households, 149 opted for this approach because it involved consuming less preferred foods that were more affordable and readily accessible in the market. During this period, these households consciously avoided culturally favored and delicious foods. In cases where the food shortage persisted, 111 households took the step of reducing the number of meals they consumed, transitioning to a regimen of eating twice a day, once a day, or occasionally skipping meals. A subsequent strategy, followed by 101 households, involved gradually reducing the portion sizes of meals served to household members.

As the available food resources continued diminishing, some households resorted to purchasing grain on credit. About sixty-nine households indicated they would try this possibility if they could obtain grain on credit. However, the likelihood of securing credit was primarily determined by their ability to command their social capital. It is distressing to see farmers consuming their seed stock. Nevertheless, fifty-four households reported resorting to this measure. In the event that the situation did not improve, households were compelled to prioritize feeding infants over adult household members. In these circumstances, fifty-one households concentrated their efforts on providing food for infants, even at the expense of adults who were capable of seeking daily wage labor and alternative sources of food. As the situation escalated from one level to the next, the food security status these households get worsened. Households that did not witness improvement in the earlier stages were compelled to resort to alternative coping mechanisms, which proved to be difficult and challenging to implement.

Diversification of crops is a significant adaptation strategy practiced by more than 267 respondents. Due to the frequency of rainfall shortages and variability and droughts, most households undertake crop diversification as the number one strategy (Figure 5). Crop diversification can take various forms, including intercropping (planting different crops together in the same field), crop rotation (changing the type of crop grown in a field from season to season), and integrating tree crops or other perennial plants into a farming system. These practices can contribute to more resilient and sustainable agricultural systems.

A popular saying in the study area is, “Livestock are insurance against crop failure.” Livestock supplies draught power for farming, manure to augment soil fertility, and a source of cash income if the need for money arises. Hence, 201 households deploy this as a survival strategy. Animal fattening, requiring the involvement of eighty-nine households, stood third among the lists in the survival strategy. According to information gathered from respondents, farmers have actively adopted the strategy of small-scale backyard fattening to boost their income. This approach involves tethering and feeding oxen, which play a crucial role in the demanding agricultural season but tend to lose weight due to their heavy workload. By nourishing these oxen, farmers ensure they reach an optimal condition and can be sold at a premium price. During the off-farm seasons, farmers replace these well-fed oxen with younger and less experienced ones, which are more affordable. This cycle is repeated as part of their farming practices.

Additionally, some household members have implemented a survival strategy where they invest their available cash in valuable and high-cost assets. In food-insecure situations, households prioritize preserving their seed stock for the upcoming farming season. This practice enables them to fulfill their seed requirements from their own stock. Out of the total households surveyed, sixty-four respondents reported implementing this strategy. Another survival strategy adopted by fifty-one households is the implementation of soil SWC practices on their individual and communal lands. Farmers understand the significance of SWC practices in preventing soil degradation and enhancing soil fertility. They recognize SWC as a crucial element for achieving sustainable food production. Eventually, households can implement more severe and critical strategies to safeguard their livelihoods and keep food security (Figure 5).

## 5. Discussion

The agricultural sector in Ethiopia holds significant importance within the economy and serves as a pivotal driver for the country’s development and achievement of food security goals. The nation’s future undeniably linked to the scale and significance of investments made in agriculture. Therefore, this chapter addresses the following mandate area of the research: Respondents’ socioeconomic characteristics, the food security situation of households, determinants of household food security, major coping and survival strategies employed by studied households and up taking the linkage between food security, land certification, and SLM.

The research identified that about 90 percent of the population belongs to age categories that show the existence of a potential workforce capable of taking part in development initiatives like watershed management. A sizeable proportion, precisely 40 percent, of the respondents lacked formal education and could not read or write. This lack of literacy posed a challenge in issuing new agricultural practices and innovations within a community primarily composed of illiterate individuals. Notably, only 42 percent of the respondents were literate, further worsening the difficulty in introducing and implementing innovative ideas related to agriculture. Other findings by [89,90] indicate that the presence of a literate farming community support and facilitate technology transfer and adoption of new farming practice, specifically the diffusion of SLM practices.

The HHFBM model has highlighted that the proportions underscore the severity of food insecurity among the households examined, indicating a significant segment grappling with heightened levels of such insecurity. A substantial portion of the households under scrutiny experience an insufficiency in kilocalories (Kcal). These Kcal units are vital for supporting the body’s essential physiological functions and activities that sustain its optimal functioning and vitality. A congruent finding was unveiled in a study conducted in central Ethiopia utilizing the HHFBM approach by Carletto et al. [78], indicated that 37.9 percent of the surveyed households were found to be facing food insecurity. Among the food-insecure households, 13.1, 9.7, 8.3, and 6.9 percent were mildly, moderately, highly, and severely food insecure [80]. Similar research performed in the adjacent watershed by Agidew and Singh [91] revealed that 79.1 percent of surveyed households found food insecure. 

Unlike HHFBM, HDDS puts more households under the food security category. Corroborating the outcomes of this investigation, a study conducted in the Western Gojam zone of the Amhara region demonstrated a favorable connection between Household Dietary Diversity Score (HDDS) and Sustainable Land Management (SLM). Engagement in SLM endeavors heightened the probability of households consuming 1.5 or more food items, equivalent to a 33 percent increase compared to those who did not embrace SLM practices [92]. The same was found in research performed in Nigeria to explore the linkage between food security and tenure security. According to the findings, 43 percent of the smallholders sampled had sufficient HDDS, while 39 percent had a moderate level [93]. As proven by Kehinde et al. [94], there is a link between HDDS, and tenure security expressed in purchasing land and the prevalence of customary land rights.

The result of MAHFP indicates that only 6.2 percent of the surveyed households can feed their household members for the entire twelve months. In research conducted using the MAHFP model in Boset Woreda of Ethiopia, the findings unveiled that 26.5 percent of the households that were surveyed encountered instances of food insecurity [88]. A substantial proportion of surveyed households showed that they could not feed their family and thus were forced to pass a vicious trend of food insecurity every year. Some households might find it challenging to sustain themselves without assistance from sources such as relatives, the Productive Safety Net Program (PSNP), or loans [95]

The onset of chronic food insecurity begins around June and persists until August or September, contingent upon the local agroecological conditions. Among these phases, chronic food insecurity is the most severe, where impoverished and middle-income households are unable to provide sustenance fully or partially for their families without external support.

Recognizing the distinct determinants of food security within specific locations empowers researchers, policymakers, and development practitioners to implement targeted interventions. With this understanding in mind, this research identified eleven determinants during a panel discussion, followed by a thorough examination of their significance levels via a structured household questionnaire. Notably, among several factors, five emerged as particularly noteworthy: insufficiency and unpredictability of rainfall, crop pest infestations, diminishing agricultural land, land degradation, and the occurrence of floods. 

Inadequate and fluctuating rainfall patterns exert detrimental impacts on both crop and livestock production. Excessive or deficient rainfall disrupts crop productivity by leading to waterlogging and drought conditions. Furthermore, heavy rainfall results in flooding, thereby inflicting harm to crops and causing soil erosion, which contributes to the degradation of agricultural plots [96]. In Africa, approximately two hundred million, or 25 percent of the population, currently experience water stress, with more countries expected to face high risks in the future. This may, in turn, lead to increased food and water insecurity for the populations [97]. Most importantly, rainfall fluctuations are experienced frequently, with some years recording high while others recording extremely low (almost zero) rainfall [98]. Additionally, the Ethiopian highlands are witnessing widespread soil and water erosion because of land degradation [99,100]. Likewise, the research conducted by Kehinde et al. [94] aligns with this conclusion, demonstrating that waterlogging has a detrimental impact on wheat, which is a primary staple crop in the surveyed region. Smallholder farmers relying on the traditional rain-fed agricultural production system experience food shortages due to erratic rainfall patterns and recurrent droughts [94]. The study conducted by Rockstrom et al. [101] revealed that the absence of sufficient rainfall and its uneven distribution in rain-fed agriculture had been frequent occurrences, leading to a significant seventy percent reduction in crop yield or even complete crop failure in some instances. Studies conducted by Gebreselassi [98] have confirmed that irregular patterns of rainfall significantly impede food production. A study conducted in Basoworana Woreda of North Shoa has shown that fluctuations in rainfall have had a detrimental impact on the food security of households. This impact includes crop failures, a reduction in available grazing land, disruptions in water supply, and increased vulnerability to spikes in food prices, as well as nutritional challenges [102]. The fluctuating patterns of rainfall and the impact of climate change on agriculture can have several adverse effects, including reduced crop yields and compromised nutritional quality [103]. Apart from the fluctuating rainfall, lack of knowledge on coping and adaptation strategies is seen as one of the causes of food insecurity in the study area because most of the farmers depend on rain for their agricultural activities [94].

Prolonged periods of excessive rainfall or drought can weaken plants, rendering them more susceptible to various pests and diseases. When there is an excess of rainfall, the soil becomes saturated, creating conducive conditions for the growth of specific pathogens, thus facilitating disease development [104].

Crop pests and diseases also affect agricultural production and food security. As reported by the respondents, Faba bean disease (Botrytis fabae), wheat rest (Puccinia gramins), and maize stalk borer (Busseola fusca) are, among others. Most specifically, the impact of Faba bean disease has been noted to be profoundly affect farmers’ livelihoods and food security, given that this staple crop serves as a significant source of both sustenance and income. Research conducted in the primary Faba-bean-growing regions of the Ethiopian highlands revealed compelling evidence. The disease’s prevalence rate was found to be as high as 85.7 percent [105], leading to substantial yield losses ranging from 50 to 62.5 percent. Furthermore, the ailment significantly impacts both the quantity and quality of the harvested beans [106]. Plant pests and diseases could potentially deprive humanity up to 82 percent of the attainable yield in the case of cotton and over 50 percent for other major crops [107] and, combined with postharvest spoilage and loss in quality, these losses become critical, especially for resource-poor small holder farmeres.

Shrinking farming plots are still the prominent hurdle in the struggle towards food security attainment. The rapid population growth, with an annual increase of 2.9 percent [108], has had a detrimental impact on the density of agricultural land in the area under study. Land, a valuable resource for the farming community, continues to diminish over time, making it insufficient to meet the food needs of farming households. According to the study by Tesfaye et al. [109], the average land size per household in the two studied Woredas declined to 0.75 hectares, significantly lower than the national average and comparable to the regional average of 1.01 and 0.7 hectares. Consequently, farmers are compelled to produce food from these tiny plots, leading to chronic and temporary food shortages for household members. The finding of this research is supported by similar research in Pakistan, which puts shortage of cropping land as one of the major determinants of agricultural production [106].

Land degradation in the study area is a phenomenon with a wide prevalence and is similarly known to have devastating effects on agricultural production. According to the discussion of this research, land degradation has a double fold on agricultural production, diminishes farm size, and reduces soil fertility and productivity [110]. Soil fertility in the study area declined due to intensive farming, overgrazing, inappropriate farming systems, and deforestation. Scholars agree on the prevalence of the problem, saying that “soil degradation is a global pandemic” [103]. Several other researchers reaffirm that it is challenging to sustain food security for a more robust livelihood in areas prone to land degradation [111,112,113,114]. Inadequate nutrient supplies, depletion of soil organic matter, and soil erosion have been major obstacles to securing the best out of the agricultural sector in Ethiopia [115]. The prevailing human-land interaction in the country has been challenging the agricultural sector, the most critical sector for food security and nutrition enhancement, growth sustenance, and poverty reduction. According to Diagana [116], soil fertility decline and nutrient mining lead to reduced agricultural productivity and, thus, food insecurity. As indicated in the report of the Food and Agricultural Organization (FAO), the problem of food and nutritional insecurity continues to increase. Its latest estimates indicate that the rate of human undernourishment has increased globally. For instance, in 2017, around one person out of every nine on earth (or 821 million people) is undernourished, 151 million under-five-year-old children are stunted, and fifty million children are threatened by wasting [117]. The same source indicates that a high proportion of undernourished, stunted, and wasted people are found in Africa, specifically in SSA. A study conducted in West Africa revealed that the proportion of children who die before the age of five years was more than 30 percent in areas of high land degradation [118]. Ethiopia is currently affected by land degradation and the consequent imbalance between the food supply and demand sides.

Households who took part in the survey have deployed several coping strategies that helped them to withstand chronic and transitory food insecurity using several measures. The implementation of each plan varies based on the degree and size of food scarcity. Hence, at the initial stage of the problem, they start by reducing the number and quality of food, like consuming from seed stock to sending household members to eat elsewhere, begging, and searching for wild fruits. A study by Tefera and Tefera [119] revealed that individuals employed various coping strategies in response to food shortages. These strategies included reducing the number and size of meals, borrowing cash and grain, receiving food aid, selling animals, taking part in food-for-work programs, engaging in off-farm and non-farm jobs, renting land, and mortgaging land. Similarly, Tolossa [120] found that coping mechanisms like change in foodstuffs and meals, support from relatives, temporary family dispersal, undertaking unusual jobs, and relying on safety nets were the type of coping strategies employed by studied households. It is uncertain to see households consuming from their seed stock. This action could be linked to the widespread problem and lack of dynamic and petite social relations. Research performed in Darfur by [76] saw communities hiding their seed stock beneath the sand, speculating about the upcoming crop season.

Adaptation entails implementing consistent responses as a long-term strategy to mitigate and avoid potential threats. Hence, surveyed households also employed various survival strategies. Among the strategies, diversification of crops, diversifying animals, animal fattening, saving money on expensive materials, seed reserve, and natural resource conservation have taken top priorities. Like most rural areas in Ethiopia, all the households in the study rely heavily on crop cultivation and livestock rearing as their primary means of sustenance. A finding of research performed by different scholars asserts related results [16,121,122,123].

Food security attainment of surveyed households is highly influenced by natural calamities that are beyond the control of the community. Moreover, whenever the trend of agricultural production remains positive with a good yield, households are unable to economically utilize the grain to jump food shortage and feed their household uninterruptedly. Nonetheless, the trends of food security and the causation of food shortage investigated in the place can be linked to the two theories of food security, FAD, and political economy explanation.

## 6. Conclusions

Food security appeared as a significant concern among the surveyed households, with a high prevalence of food insecurity trends seen across different models. Most households fell into low and medium dietary diversity categories, showing limited access to various food items. Notably, only a tiny percentage of households could supply food for their members throughout the year, highlighting the severity of food insecurity.

Various determinants of food security were found, including inadequate rainfall and its variability, crop pests and diseases, diminishing farmland, land degradation, and floods. These factors had adverse effects on agricultural production and the availability of food. Although interventions such as land certification and SLM were implemented, their impact on household food security has not yet been pronounced. The prevailing issue of extensive land degradation posed a significant obstacle to achieving current food security goals. The naturally occurring determinants, such as rainfall variability, crop pests and diseases, and shrinking land size, were the primary contributors to food security challenges in the study area.

Households have adopted various coping strategies to address food insecurity, such as reducing the quantity and quality of meals, borrowing cash and grain, relying on food aid and safety nets, and seeking off-farm and non-farm employment opportunities. Adaptation strategies focused on diversifying crops and livestock, engaging in animal fattening, and conserving natural resources.

## 7. Recommendations

Despite a huge intervention in land management and certification programs, the food security situation of studied households is in a precarious situation. Albeit changes were made in the intervention and results obtained in terms of preventing land degradation and boosting tenure security, the naturally occurring determinants have outweighed the outcome of these two interventions. Addressing food security challenges in the study area requires comprehensive and integrated approaches. Based on the findings from the analysis, the following policy implications become known for improving rural household food security in the study area as well as in Ethiopia:The Government of Ethiopia should make the necessary amendments to the existing land tenure system to provide individual, communal, and public land rights and introduce a cadaster system.The government should invest in education and training to improve the capacity, knowledge, and skills of the farmers to ensure equal opportunities and to improve income, agricultural production, and sustenance of food security.The Ethiopian government and local institutions should enhance the current Soil and Water Conservation (SWC) program in a manner that actively engages the local community and incorporates cultural and religious perspectives. This can be achieved by making full use of the existing laws and bylaws.To ensure the sustainable and effective use of produced grain, the local government needs to be aware and train the community on the proper utilization, consumption, storage, and trade of grain.The local government should sensitize farmers and avail opportunities in off-farm and non-farm activities. These income-generating activities should be linked to the sustainable use of natural resources, livelihood assets, and income to increase agricultural productivity and food security.The local government and other stakeholders should be aware of and train farmers on climate-smart agriculture in such a way that addresses climate change’s impact on agriculture while simultaneously improving productivity, sustainability, and the well-being of farmers and communities.Local governments should encourage diversification in both crop and livestock farming. This can be achieved by ensuring that grazing fields are managed to maintain their carrying capacity, preventing overgrazing that can lead to environmental degradation.

## Figures and Tables

**Figure 1 foods-12-03341-f001:**
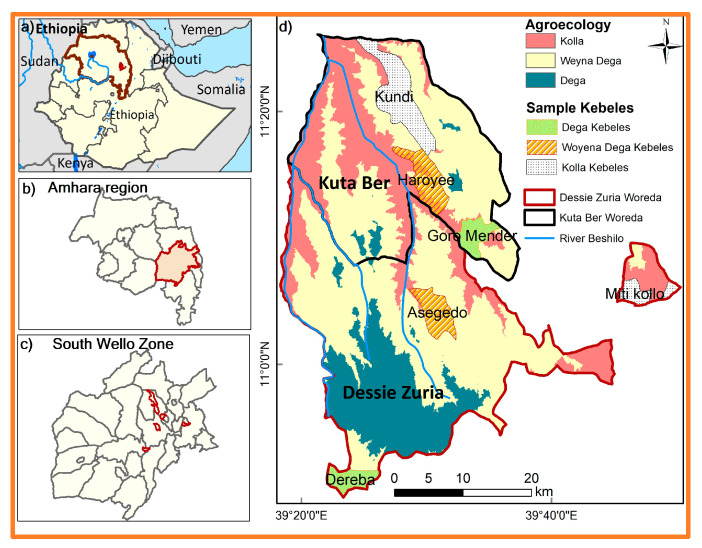
Location map of the study area: (**a**) Ethiopia; (**b**) Amhara region; (**c**) South Wello Zone; and (**d**) Study Woredas.

**Figure 2 foods-12-03341-f002:**
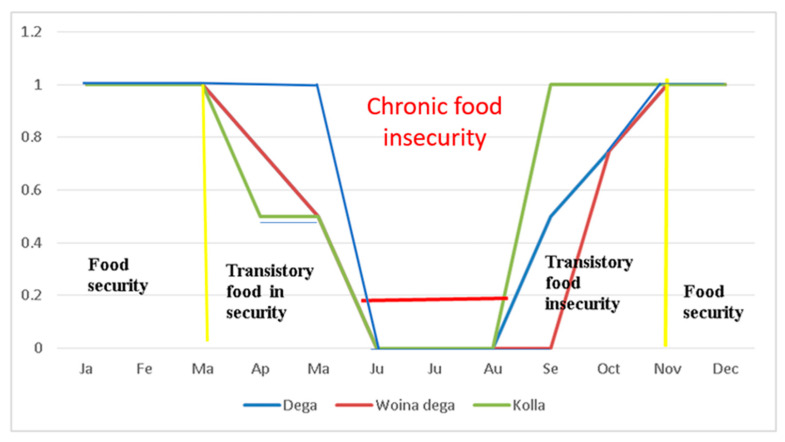
Patterns of food security and insecurity across various agroecological zones.

**Figure 3 foods-12-03341-f003:**
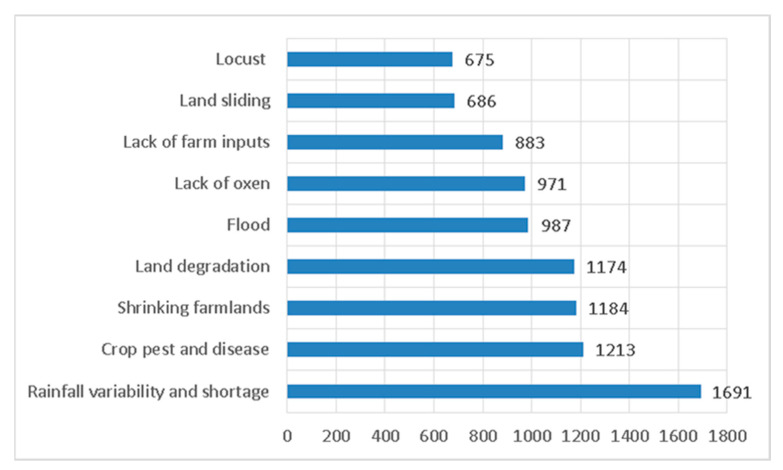
Determinants of household food security.

**Figure 4 foods-12-03341-f004:**
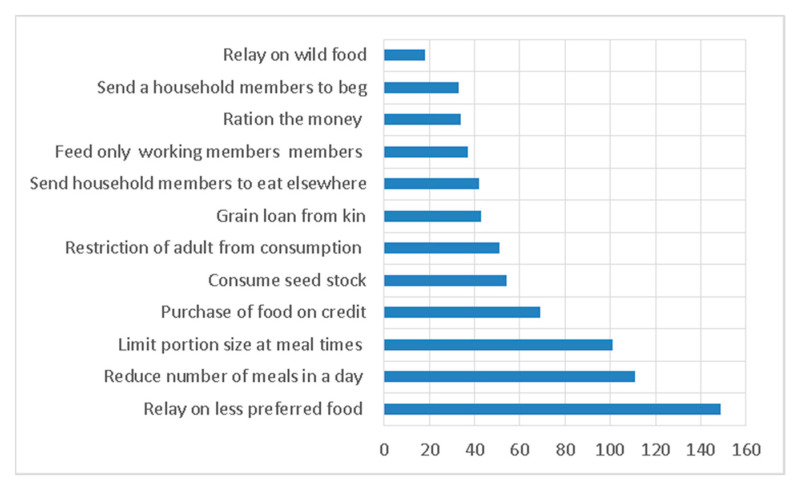
Number of households using a coping strategy.

**Figure 5 foods-12-03341-f005:**
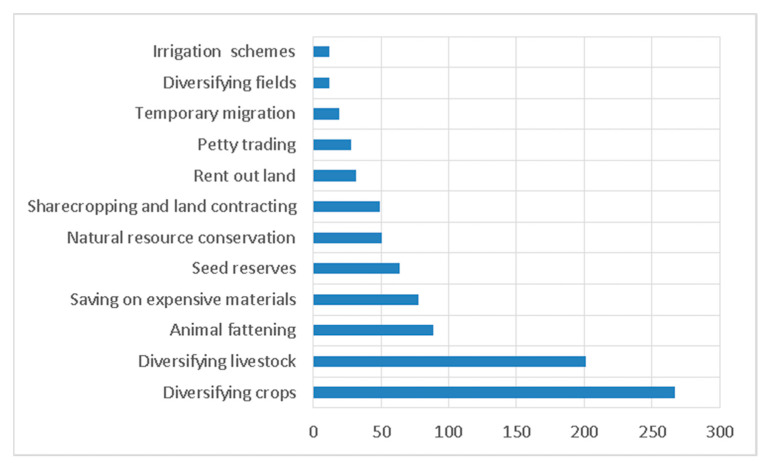
Number of households using survival strategies.

**Table 1 foods-12-03341-t001:** Data type and collection methods using agroecology.

Woredas	Dessie Zuria			Kutaber			Total
Kebele	Dereba	Asgedo	Mitikollo	Haroyee	Goromender	Kundi	6
Agroecology	Dega	Woyenadega	Kolla	Dega	Woyenadega	Kolla	3
Households (N)	632	1100	686	731	1038	2045	6232
Survey households (n)	38	65	41	44	62	121	371
KII	3	5	3	3	5	6	25
FGD	2	2	2	2	2	2	12
Community workshop	1	1	1	1	1	1	6

**Table 2 foods-12-03341-t002:** Food security status of studied households based on the HHFBM model.

Household Food Security Status		Per Capita Dietary Kcal Available	Freq.	Percent	Cum.
Severely food insecure		≤1500	259	69.65	69.65
Moderately food insecure		≥1500–1750	42	11.45	81.09
Mildly food insecure		≥1750–2100	27	7.2	88.31
Food secured		≥2100	43	11.7	100.00
Total			371	100.00	
Min (Kcal)	Max (Kcal)	Mean (Kcal)		Std (Kcal)	
78.1	7547.3	1272.1		867.5	

**Table 3 foods-12-03341-t003:** The food security situation of households is based on the HDDS model.

Food Security Category	Number of Households Falling in the Category	Percent of Households Falling in the Category	CUM.
High DD	239	64.4	64.4
Medium DD	45	12.2	76.6
Low DD	87	23.4	100
Total	371	100	-

**Table 4 foods-12-03341-t004:** Food security situation of households based on MAHFP model.

Food Security Status Category	Months of Adequate Food Provisioning	Number of Households Falling in the Respective Category	Percent of Households Falling in Respective Food Security Category	Cum
Food Secure	≥12	23	6.2	6.2
Mildly food insecure	≥10–12	113	30.4	36.6
Moderately food insecure	≥7–9	166	44.8	81.4
Severely food insecure	≤6	69	18.6	100
Total		371	100	

## Data Availability

The data will be made accessible through the University of Twente’s DANS (Data Archiving and Networking Service).

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
