# Peer review of "A Holistic Analysis of Food Security Situation of Households Engaged in Land Certification and Sustainable Land Management Programs: South Wello, Ethiopia"

_foods, 2023, doi:10.3390/foods12183341_

Round 1
Reviewer 1 Report
This is an interesting and informative study of an important topic. The presentation of the study could be improved by minimizing the repetition of key points. Repetition can be useful for purposes of emphasis but can also add unnecessarily to the length and readability of the article. For example, the "Discussion" is basically a repeat of the "Results" rather than an elaboration or discussion of implications.
Reviewer 2 Report
Thank you for writing interesting paper. However, there are some rooms to improve.
1. The Government of Ethiopia (GoE) has launched large-scale land certification and sustainable land management programs to improve agriculture.Does the program only target large-scale land? How about the small and medium-scale land?
2. The way author wrote objective in line 79-82 need to revise it more neatly
3. What is the research gap in this study?
4. The author should use new concept of Food Security, not from 1996 World Food Summit
5. The literature review is tedious. Please explain it more briefly.
6. Does the Figure 1 has no copyright issue?
7. The Data Analysis Technique should be explained more detail. Is there no econometrics analysis used?
8. Please add sub section of Agriculture Overview for each area of study that represent the different agroecology.
9. There is no innovation in the methodology. Please add empirical model by using econometrics
Reviewer 3 Report
Journal
Foods (ISSN 2304-8158)
Manuscript ID
foods-2561035
Type
Article
Title
A Holistic Analysis of Food Security Situation of Households Engaged in Land Certification and Sustainable Land Management: South Wello, Ethiopia
The presented study addresses a concern, highlighting land degradation as a prominent issue in the northern highlands of Ethiopia, which is recognized as the country's famine corridor.
It acknowledges the Ethiopian government's initiatives in sustainable land management and certification programs aimed at mitigating challenges related to food security, land management, and tenure security since 2000.
The study zeroes in on households engaged in these programs, delving into their food security status, underlying determinants, and coping strategies. To comprehensively analyze this multifaceted issue, a combination of primary and secondary data from qualitative and quantitative sources was employed.
Quantitative data from surveyed households were subjected to statistical analysis, while qualitative data underwent transcription, categorization, and interpretation. The study evaluated food security utilizing three models: the Household Food Balance Model (HHFBM), Months of Adequate Household Food Provisioning (MAHFP), and Household Dietary Diversity Score (HDDS). Strikingly, a significant proportion of surveyed households, comprising 88.3%, 35.6%, and 93.8% respectively, demonstrated food insecurity based on these models.
Evidently, the research identifies some of the key determinants of food security, including factors such as rainfall scarcity and variability, crop-related challenges such as pests and diseases, diminishing farm sizes, and land degradation. However, more dwelling into factors such as COVID-19 is missing (i.e.: https://globalizationandhealth.biomedcentral.com/articles/10.1186/s12992-023-00952-7 and https://doi.org/10.1016/j.foodpol.2020.102017 and https://doi.org/10.1016/j.spc.2022.01.007).
While the study sheds light on the pressing issue of food security and land degradation in northern Ethiopia, further inquiry need to delve into the interplay between these factors and their broader socio-economic and environmental implications.
Definitely the research tackles a crucial topic, focusing on land degradation in northern Ethiopia's highlands. While the intent is commendable, the execution of the study raises still certain concerns.
The methodological approach, as mentioned earlier combines qualitative and quantitative data collection, seems promising. However, the lack of detailed information about the sampling strategy, survey design, and data collection process leaves a gap in understanding the representativeness and reliability of the findings. The reliance on SPSS software for statistical analysis is acknowledged, but a more detailed description of the statistical methods employed would have bolstered the study's rigor. For this type of study is not enough to have a brief description of “Data analysis technique”.
Although the study employs established food security assessment models, such as the Household Food Balance Model and Dietary Diversity Score, the results reporting lacks depth. This can be however enhanced in Discussion.
The research identifies crucial determinants of food security, such as rainfall patterns, crop challenges, and land degradation. Yet, the study's analysis of these determinants remains superficial, missing an opportunity for deeper exploration and correlation with the observed food security outcomes.
Furthermore, the study is missing some clear set of recommendations. At the moment, it is rather providing set of overviews from other sources. I.E. while the study recommends a series of reforms and interventions, the feasibility and potential challenges associated with these proposals are not thoroughly examined and justified. A critical assessment of the institutional, economic, and social barriers to implementing these recommendations would have added practicality to the study's implications.
In conclusion, the research addresses a significant issue but falls short in providing a comprehensive and robust analysis. Strengthening the methodology, deepening the analysis of determinants, and critically assessing the proposed recommendations would enhance the study's contribution to the understanding of food security and land degradation in northern Ethiopia's highlands.
Additional comments:
-language needs to be improved
-I do not understand, why the manuscript is not in the template
-some of the recent papers on the topic in Ethiopia should be also discussed, i.e.: https://agricultureandfoodsecurity.biomedcentral.com/articles/10.1186/s40066-022-00381-6 and https://doi.org/10.1016/j.cliser.2022.100307 and maybe https://www.mdpi.com/2673-3986/3/2/13 (last one linked to the earlier comment of missing effects of COVID-19)
-finally, I am not a big fan of the “Related literature” section. I would prefer to have it incorporated in a concise way in Introduction. But I am leaving this up to the authors.
With regards,
The Reviewer
As indicated earlier in the "review system", therefore I do not understand why is here this "obligatory" section.
Round 2
Reviewer 2 Report
Thank you for the largely revised. Please do not use bullet for the objective of the study.
Reviewer 3 Report
Dear Authors,
I can see significant improvements in the work.
However, some of my comments are still partly remaining.
For example:
Point nine: I do not understand, why the manuscript is not in the template
Response nine : Thank you. We are sorry for not using the template of the manuscript.
This is truly a response, but not a satisfactory one. In the review stage, I would like to see it sorted out.
Point seven: Furthermore, the study is missing some clear set of recommendations. At the moment, it is rather providing set of overviews from other sources. I.E. while the study recommends a series of reforms and interventions, the feasibility and potential challenges associated with these proposals are not thoroughly examined and justified. A critical assessment of the institutional, economic, and social barriers to implementing these recommendations would have added practicality to the study's.
Response seven: We thank you very much again. Indeed, you are right and pretty correct about the addition of proper recommendations pertinent to the scope of the study. Based on your comment, we have cordially put recommendation suiting to the finding of the research. See highlighted in green)
This still needs to be improved. I recommend to logical place it into a separate chapter. Which will help you to have a clear and practical recommendations.
Point six: The research identifies crucial determinants of food security, such as rainfall patterns, crop challenges, and land degradation. Yet, the study's analysis of these determinants remains superficial, missing an opportunity for deeper exploration and correlation with the observed food security outcomes.
Response six: Thank you again. The point is so interesting and worth mentioning. We think that, this point has similarity with point three, where we tried to address that particular question. Nonetheless, we thank you for that.
The response needs to be more concise and specific. And same for the manuscript.
Point two: Evidently, the research identifies some of the key determinants of food security, including factors such as rainfall scarcity and variability, crop-related challenges such as pests and diseases, diminishing farm sizes, and land degradation. However, more dwelling into factors such as COVID-19 is missing (i.e.: https://globalizationandhealth.biomedcentral.com/articles/10.1186/s12992-023-00952-7 and https://doi.org/10.1016/j.foodpol.2020.102017 and https://doi.org/10.1016/j.spc.2022.01.007).
Response two: Indeed, your observation is highly pertinent, and we deeply appreciate your insight. It's evident that the COVID-19 pandemic has had a profound and detrimental impact on various aspects of agricultural production. The disruption it caused across the three fundamental pillars of production - namely, production, distribution, and consumption - is undeniable. In alignment with your valuable suggestion, we took the initiative to review the article titled "Viewpoint: characterizing agricultural and policy response to outbreak of COVID-19." This piece of work is truly exemplary in its analysis of the agricultural and policy responses triggered by the COVID-19 outbreak. However, we must acknowledge that the constraints imposed by the limited scope, dictated by both time and budget considerations, hindered our ability to comprehensively examine the pandemic's influence on the food security landscape of farming households. As indicated in the introduction section, which is helpfully highlighted in green, we made a deliberate effort to reference significant works that elucidate the global effects of the pandemic. Thank you for recognizing this aspect, and we genuinely appreciate your understanding of the challenges we encountered within the scope of this study.
I recommend strengthening that part by two remaining studies as well.
Finally, in addition, some of the responses seem to be AI generated. Which I consider as a significant problem, which raises questions. Luckilly, I do not have that feeling from the manuscript, so I will let it slide for now. But I expect more coherent improvements.
N/A
